# Distinct Changes in Calpain and Calpastatin during PNS Myelination and Demyelination in Rodent Models

**DOI:** 10.3390/ijms232315443

**Published:** 2022-12-06

**Authors:** John A. Miller, Domenica E. Drouet, Leonid M. Yermakov, Mahmoud S. Elbasiouny, Fatima Z. Bensabeur, Michael Bottomley, Keiichiro Susuki

**Affiliations:** 1Department of Neuroscience, Cell Biology, and Physiology, Boonshoft School of Medicine, Wright State University, Dayton, OH 45435, USA; 2Department of Mathematics and Statistics, Wright State University, Dayton, OH 45435, USA

**Keywords:** calpain, calpastatin, myelination, demyelination, peripheral nerve

## Abstract

Myelin forming around axons provides electrical insulation and ensures rapid and efficient transmission of electrical impulses. Disruptions to myelinated nerves often result in nerve conduction failure along with neurological symptoms and long-term disability. In the central nervous system, calpains, a family of calcium dependent cysteine proteases, have been shown to have a role in developmental myelination and in demyelinating diseases. The roles of calpains in myelination and demyelination in the peripheral nervous system remain unclear. Here, we show a transient increase of activated CAPN1, a major calpain isoform, in postnatal rat sciatic nerves when myelin is actively formed. Expression of the endogenous calpain inhibitor, calpastatin, showed a steady decrease throughout the period of peripheral nerve development. In the sciatic nerves of *Trembler-J* mice characterized by dysmyelination, expression levels of CAPN1 and calpastatin and calpain activity were significantly increased. In lysolecithin-induced acute demyelination in adult rat sciatic nerves, we show an increase of CAPN1 and decrease of calpastatin expression. These changes in the calpain-calpastatin system are distinct from those during central nervous system development or in acute axonal degeneration in peripheral nerves. Our results suggest that the calpain-calpastatin system has putative roles in myelination and demyelinating diseases of peripheral nerves.

## 1. Introduction

To ensure rapid and efficient nerve conduction, neurons and glial cells form intricate structural and functional connections to optimize the transmission of action potentials over long distances [1,2]. Myelinating glial cells enwrap axons with myelin that serves as electrical insulation for axons and aids in the formation of specialized axonal domains, both of which assist in saltatory conduction [2]. The importance of myelin is further underscored by diseases involving the disruption of myelinated nerve fibers in the peripheral nervous system (PNS). Peripheral neuropathies involving myelinated structures are notoriously difficult to treat and often cause severe weakness, pain, and long-term disability [3]. However, detailed mechanisms of myelination and demyelination in the PNS remain unclear.

Calpains, a family of calcium dependent cysteine proteases, have been shown to be key in the myelination and demyelination process in the central nervous system (CNS). Calpains mediate many aspects of cellular degradation and cytoskeletal rearrangement [4]. Calpain activity in the cell is tightly regulated with calcium acting as an activator and calpastatin acting as a calpain-specific endogenous inhibitor [4]. Calpain-1 and Calpain-2 are conventional/classical calpains in that they are found ubiquitously, contain a penta-EF hand domain, and are associated with a small regulatory subunit. Calpain nomenclature refers to the heterodimer of the large catalytic subunit (CAPN1 and CAPN2) and the common small regulatory subunit (CAPNS1) [5]. CAPN1 (or μ-calpain) is activated at μM calcium concentrations, while CAPN2 (or m-calpain) has mM calcium concentration requirements [4]. Previous studies have characterized developmental regulation of the expression of calpains and calpastatin in the CNS [6,7]. In oligodendrocytes, the myelinating glial cells of the CNS, increased calcium and calpain levels are necessary to retract initially over-produced myelin sheaths during development [8]. Retraction of myelin sheaths in this dynamic time of development appears to be an activity-dependent process ensuring proper CNS structure and function [8]. From an axonal perspective, increased axonal calcium and calpain has also been shown to cause myelin to retract from the paranodes in response to high-frequency stimulation [9]. Furthermore, calpains have well-established roles in the hallmark features of demyelination and neurodegeneration seen in the CNS demyelinating disease multiple sclerosis [10]. These studies establish a role for calpain during the maturation of myelin and demyelination in the CNS.

Calpains have also been implicated in axonal pathology in the PNS. After sciatic nerve transection, calpains have been shown to cause granular disintegration of the axonal cytoskeleton after a transection injury [11,12]. Collapsin response mediator protein 4, a regulator of the actin and microtubule cytoskeleton, is cleaved by calpains, and its fragments promote axonal degeneration in the PNS [13], suggesting a potential mechanistic link between calpains and axonal degeneration. Interestingly, the overexpression of calpastatin, or its enhanced axonal transport, delayed axonal degeneration and neuromuscular synaptic loss following transection of the sciatic nerve [14]. Calpains contribute to the node of Ranvier disruption of the axonal form of Guillain-Barré syndrome [15]. In neuropathic pain models, calpains have been implicated in the proteolysis of myelin associated proteins and inhibiting calpains prevented the degradation of these proteins and pain-like behaviors [16,17]. However, the role of calpain in primarily demyelinating conditions in the PNS has not been well-explored. It has been suggested that calpains have putative roles, distinct from the CNS, in the maintenance of peripheral nerve structures, since calpain-specific αII spectrin breakdown products [18] are absent in the normal adult rat brain but are found in normal adult peripheral nerves [19]. Therefore, it is important to elucidate the PNS-specific roles of the calpain-calpastatin system.

As a first step in examining calpains’ potential role in peripheral demyelinating diseases, we here characterize the calpains and calpastatin during postnatal development, in a chronic model when normal myelination is disrupted, and in an acute demyelinating model. During the early developmental phase, levels of activated CAPN1 and calpastatin were high compared to adult levels in rat sciatic nerves, which is distinct from CNS development. In a well-established chronic dysmyelinating model, we found increases in both CAPN1 and calpastatin. In adult rodents using a well-established acute demyelinating model, we found a significant increase in the expression of CAPN1 and a decrease in calpastatin, which is distinct from an acute axonal degeneration model. Together, our results suggest that the calpain-calpastatin system is associated with the maturation and disruption of myelin in the PNS.

## 2. Results

### 2.1. Calpain-Calpastatin Expression Is Developmentally Regulated in the PNS

In the calpain family, CAPN1 and CAPN2 are ubiquitously expressed and have been shown to have putative, and potentially opposing, roles in normal development and diseases in the CNS [20]. Previous studies showed that the expression of these calpains and their endogenous inhibitor, calpastatin, is developmentally regulated in the CNS [6,7], consistent with the key roles of calpains in CNS myelin development [8]. Although calpains have been shown to cause axonal degeneration in response to injury in the PNS [11,12], the expression patterns and roles of the calpains and calpastatin during PNS development are less clear. In the rodent PNS, myelination begins around birth, peaks around 2–3 weeks after birth, and then decreases to a low basal level around the end of the first month [21]. In recent studies examining the transcript levels, postnatal day (P) 6 was shown to be the peak of many myelin components and described as early myelin biogenesis [22]. Another group of myelin transcripts were co-elevated at P18, corresponding to a later phase of myelination [22]. For example, transcripts of dystrophin-related protein 2 (DRP2), a well-established component of PNS myelin [23], were shown to peak at P18 [22]. Using bulk RNA sequencing of the sciatic nerve, DRP2 transcripts have been shown to be increasing throughout postnatal development [24]. Consistent with these results, the expression of the myelin protein DRP2 increased over the first 3–4 weeks postnatally during the expected developmental myelination phase in the PNS (Figure 1A). To determine the regulation of calpains and calpastatin in the PNS during development, we examined the expression of these proteins in sciatic nerve homogenates postnatally from Sprague Dawley (SD) rats. The expression levels of full-length CAPN1 (80 kDa) were relatively constant throughout the development of rat sciatic nerves (Figure 1A,B). These results contrast the increase of CAPN1 expression throughout CNS development to adult levels [7]. Interestingly, immunoreactivity of the amino-processed (76 kDa and 78 kDa), or previously described as activated forms of CAPN1 [25,26], showed an increase between P16 and P28 and decrease in adults to nearly the level at P0 (Figure 1A,B). We also saw an increase of CAPN1 immunoreactivity at 68 kDa (Figure 1A), which has previously been described as a cleaved form of CAPN1 [27], although whether this cleaved CAPN1 is related to an increase in activity is not yet known. These data suggest that CAPN1 is activated when myelin is actively formed during PNS development, while the expression levels of full-length CAPN1 (80 kDa) are relatively maintained. The expression levels of CAPN2 did not show dramatic changes during PNS development (Figure 1A,C), similar to rat CNS [7]. Immunoreactivity of calpastatin revealed double bands around 120 kDa that are likely a result of alternative splicing events of the calpastatin gene [4]. We quantified the density of both bands together, since these different calpastatin polypeptides have shown similar interactions with calpains [4]. The expression levels of calpastatin greatly decreased throughout development to adult levels in the PNS (Figure 1A,D). This pattern directly contrasts the previously shown postnatal increase of protein and mRNA expression of calpastatin in rat CNS [6,7]. Overall, these results highlight the distinction of calpain and calpastatin regulation during development in the PNS and CNS.

### 2.2. CAPN1 and Calpastatin Expression Increases in Trembler-J Mouse Model of CMT1E

We next aimed to determine the expression of calpains and calpastatin when normal PNS myelination is disrupted. *Trembler-J* (*Tr-J*) mice carrying a spontaneous L16P mutation in the peripheral myelin protein 22 (PMP-22) are a well-established, clinically relevant model for Charcot-Marie-Tooth Disease Type 1E (CMT1E) [28,29]. Pathologically, the *Tr-J* phenotype of dysmyelination is expressed during early myelin formation (P4–P12), displaying an increased proportion of fibers that are incompletely surrounded by Schwann cell cytoplasm, and appears more severe in older mice [30,31]. Adult *Tr-J* mice display tremors while walking, slowed nerve conduction, and histologically display fewer myelinated fibers, abnormally thin myelin, and altered axonal ion channels at nodes of Ranvier and juxtaparanodes [28,30,31,32]. *Tr-J* mice show slowly progressive distal axonal degeneration indicated by denervated neuromuscular junctions and cutaneous nerve fiber loss over many months, although only mild signs of axonal degeneration are observed in the proximal sciatic nerves [33]. To examine the calpain-calpastatin system in the inherited neuropathy with dysmyelination, we collected sciatic nerves from wild-type (WT) and *Tr-J* mice at 8 weeks of age. At first, to characterize the myelin defects, we examined myelin, axon, and nodes of Ranvier structures in the sciatic nerves of WT and *Tr-J* mice. In teased *Tr-J* sciatic nerves, 60% of nerve fibers displayed disorganized localization of the myelin protein DRP2 and another 33% had an apparent lack of DRP2 staining (*n* = 4 with about 100 myelin sheaths examined per animal) (Figure 2A–D). Neurofascin localized in Schmidt-Lanterman incisures in WT (Figure 2A), but it showed diffuse staining in *Tr-J* sciatic nerves (Figure 2B). Disorganized DRP2 staining was associated with the axons with preserved neurofilament-M (NF-M) staining (Figure 2D). Consistent with the increased number of Schwann cell nuclei [30] and inappropriately short internodal length [32] in *Tr-J* sciatic nerves, we found a dramatic increase in the number of nodes of Ranvier per field of view in *Tr-J* mice (*p* < 0.0001) (Figure 2E–G). As previously reported [32], the vast majority of nodes and paranodes in *Tr-J* mice displayed some degree of disorganization due to dysmyelination (Figure 2H).

Western blotting of sciatic nerve homogenates from *Tr-J* mice showed significant increases in both CAPN1 (*p* = 0.0187) (Figure 3A,B) and calpastatin (*p* = 0.0487) (Figure 3A,D) when compared with WT mice. In mouse sciatic nerve homogenates, the full-length and amino-processed (activated) forms of CAPN1 did not separate well, so these were all quantified together. CAPN2 tended to decrease in *Tr-J* mouse sciatic nerves, although the difference did not reach statistical significance (*p* = 0.0789) (Figure 3A,C). Since there were increases in both CAPN1 and calpastatin, it is difficult to predict the net effect on calpain activity. Therefore, we next examined calpain activity with a calpain specific substrate cleavage assay using 3 sciatic nerve samples in each group available for this assay. A low sample size would decrease the statistical power and increase the possibility of Type II error in failing to reject null hypothesis. Nevertheless, calpain activity was shown to be significantly increased in sciatic nerve homogenates from *Tr-J* mice when compared with WT mice (*p* = 0.0049) (Figure 3E). These results strongly support the idea that the calpain-calpastatin system is involved in the dysmyelination of the PNS.

### 2.3. Calpain and Calpastatin Expression during Acute Focal Demyelination in PNS

In the CNS demyelinating disease of multiple sclerosis, expression of calpain and the calpain-specific cleavage product of fodrin (αII spectrin) are increased in demyelinating plaques of human patients [10]. In the experimental allergic encephalomyelitis, an animal model of multiple sclerosis, calpains were shown to be causal in the breakdown of myelin proteins [34]. In the PNS, calpains can cause axonal degeneration [11,12], however, it is unknown whether calpains are involved in acquired demyelination of the PNS. To determine the changes in the calpain-calpastatin system during acute demyelination, lysolecithin (L-α-Lysophosphatidylcholine) was injected into the sciatic nerve of adult SD rats. In this model, lysolecithin causes extensive demyelination and lysis of Schwann cells focally at the area of injection within 1 day, reaching a peak about one week after injection, while damage to the myelinated axons has been reported to be minimal [35,36]. Consistent with these previous studies, we found disrupted myelin (DRP2) but preserved axons (NF-M) 6 days after lysolecithin injection (Figure 4A,B). To further examine the effects of the intraneural injection (at the site between the knee and sciatic notch) of lysolecithin on the axons, we examined the tibial nerve segment between the knee and ankle, distal to the site of each the vehicle injection, the lysolecithin injection, and the crush injury. We used immunostaining for nodes of Ranvier as an indicator of axonal degeneration that extends distally from the injury sites [37]. Six days after a crush sciatic nerve injury causing primary axonal degeneration, there was a complete loss of nodal structures (Figure 4F,G) distally, as previously reported [37]. Intact sciatic nerves and vehicle injected nerves had very similar numbers of nodes of Ranvier in the distal tibial sections (*p* = 0.9561) (Figure 4C,E,G) demonstrating that the insertion of the needle and volume of injection had little to no effect on axonal structures. Unexpectedly, there was a reduction in the number of nodes of Ranvier structures in the tibial nerve sections distal to a lysolecithin-induced demyelination when compared with the intact nerves (*p* = 0.0041) and vehicle injected nerves (*p* = 0.0023) (Figure 4C–E,G). However, western blotting showed no difference in the axonal cytoskeletal protein NF-M between vehicle and lysolecithin injected nerves at the site of injection (*p* = 0.4437) (Figure 5A,E), despite a dramatic reduction in the myelin protein DRP2 (*p* < 0.0001) (Figure 5A,F). These results suggest that extensive demyelination due to a relatively large amount of lysolecithin caused mild damage to myelinated axons. In contrast, in the crush injury model, there were dramatic reductions in both NF-M (*p* < 0.0001) (Figure 6A,E) and DRP2 (*p* < 0.0001) (Figure 6A,F) distal to the site of the injury. Taken together, these findings confirm that lysolecithin-induced sciatic nerve injury caused a focal primary demyelination [35,36], distinct from demyelination secondary to axonal damage such as a sciatic nerve crush injury [38].

Using the lysolecithin demyelinating model, we compared the protein expression of calpains and calpastatin 6 days after either a vehicle or lysolecithin injection, around the nadir of demyelination. Full length CAPN1 (80 kDa) (*p* = 0.0446) and the activated form (76/78 kDa) (*p* = 0.0021) were both significantly increased during demyelination (Figure 5A,B). There was no change found in CAPN2 (*p* = 0.3745) (Figure 5A,C). Interestingly, there was a significant reduction in calpastatin expression (*p* = 0.0003), suggesting the increase of overall calpain activities. These results suggest that the calpain-calpastatin system is involved in the pathophysiology of acute demyelination in PNS.

### 2.4. Calpain and Calpastatin Expression during Acute Axonal Degeneration in PNS

Since we did see some potential axonal degeneration distal to the lysolecithin injection site (Figure 4C–G), we next determined the expression of calpains and calpastatin 6 days after a crush injury of the sciatic nerve, which would cause primary axonal degeneration and secondary demyelination (Figure 4 and Figure 6A,E,F). Here, we found an increase in activated CAPN1 in the crushed nerves (*p* = 0.0008), similar to the lysolecithin-induced demyelination (Figure 5A,B). However, no change was observed in full length CAPN1 (*p* = 0.5637) in crushed nerves (Figure 6A,B), whereas it was increased in the lysolecithin-induced demyelination (Figure 5A,B). CAPN2 was increased in the crushed nerves (*p* = 0.0078) (Figure 6A,C), whereas there was no change in the lysolecithin-induced demyelination (Figure 5A,C). Finally, calpastatin expression in intact nerves and crush-injured nerves was comparable (*p* = 0.1654) (Figure 6A,D), whereas it was decreased during lysolecithin-induced demyelination (Figure 5A,D). These results indicate that the changes in calpains and calpastatin in lysolecithin-induced demyelination (Figure 5) are not due to mild axonal degeneration (Figure 4 and Figure 5) and highlight the distinct changes of the calpains and calpastatin during acute primary demyelination.

## 3. Discussion 

Previous studies indicate that calpains play key roles in developmental myelination and demyelinating diseases in the CNS [8,10]. However, calpains have not been well studied in myelination and demyelination in the PNS. Our results here show that developmental regulation of calpains and calpastatin in the PNS is distinct from that in the CNS. In sciatic nerves from *Tr-J* mice with chronic dysmyelination, we show an increase in calpain activity and CAPN1 and calpastatin expression. When examining acute demyelination after lysolecithin injection and acute axonal degeneration after sciatic nerve crush, we demonstrate distinct changes in calpains and calpastatin expression. Taken together, our findings suggest unique roles of calpains and calpastatin during the time of PNS developmental myelination as well as in the pathophysiology of a wide variety of PNS diseases characterized by myelin defects.

### 3.1. Calpains and Calpastatin during Development in the PNS vs. CNS

Our results highlight the difference of the calpain-calpastatin system between the PNS and CNS. This is not surprising, since there are different mechanisms dictating the myelin development and maintenance between the PNS and CNS [39]. Regulation of calpains and calpastatin during development in the CNS is controversial. In rat brain, one study showed that CAPN1 and calpastatin gene and protein expression steadily increased, while CAPN2 levels remained relatively constant from birth to P90 [7]. Another study in mouse brain reported no change in Calpain-1 and a slight decline in Calpain-2 protein levels between P5 and P60 [40], which conflicted a report showing a postnatal decline in Calpain-1 activity and postnatal increase in Calpain-2 activity [41]. High levels of calpain activity throughout development in the hindbrain was also reported, while activity dropped sharply during postnatal development in the forebrain [42], which would be consistent with other studies showing increases in calpastatin expression [7] and activity [6] in the CNS. The reasons for these discrepancies in the CNS are not clear.

In contrast to the findings in the CNS, our results during development of the PNS showed that full-length CAPN1 and CAPN2 protein levels remained relatively constant in the rat sciatic nerves throughout development (Figure 1). However, the activated form of CAPN1 (76/78 kDa) showed a sharp increase, peaking between P16 and P28, then it subsequently decreased in adults to nearly the level at P0 (Figure 1A,B). The striking feature in the PNS is that the endogenous calpain-specific inhibitor calpastatin displayed a remarkable decline in expression throughout development (Figure 1A,D). This contrasts with the postnatal increase from initially low levels of calpastatin gene and protein expression in rat CNS [7] and calpastatin activity in rabbit CNS [6]. Since expression levels of both CAPN1 and CAPN2 in adults were similar to those at P0 (Figure 1B,C), decreased calpastatin levels might lead to the increase of net calpain activity in adult peripheral nerves compared to early development. These results suggest that a basal calpain activity is required beyond development in the PNS, presumably in the maintenance of PNS structures including myelinated axons. Consistent with this idea, calpain-specific breakdown products of αII spectrin are found in normal adult rat sciatic nerves but not in normal brain tissue [19].

The transient increase of the activated form of CAPN1 (76/78 kDa) in the PNS is observed during the period when myelin is actively formed (Figure 1A), suggesting that CAPN1 plays important roles in myelination by Schwann cells. Recent RNA-sequencing and proteome studies support this idea. Using bulk and single cell RNA sequencing of peripheral nerves and Schwann cells [The transcriptome resource Sciatic Nerve Atlas (SNAT) https://www.snat.ethz.ch], CAPN1 and CAPN2 transcripts increased during development [24]. In a separate study using RNA sequencing of peripheral myelin, CAPN1 and CAPN2 transcripts peaked at P18, coinciding with known myelin transcripts such as DRP2 [22]. Previous proteomics analysis of mouse peripheral myelin revealed expression of CAPN1, CAPN2 and calpastatin at P21, although a time course analysis was not performed [22]. In the mature PNS, protein expression of CAPN2 was found in myelin [43] and Schwann cell cytoplasm [44], but was also found in the axoplasm [44]. Calpains may also play important roles in the myelinated axons during development. For example, the increase of internodal length matches the increase of the peripheral nerve length (i.e., growth rate of limbs) between P2 and P40 [45]. Therefore, in addition to the myelin formation and elongation, activated calpains may be involved in remodeling of specialized domains in the myelinated axons (e.g., nodes of Ranvier) in response to the increase of internodal length. Similar to the transient increase of the activated form of CAPN1 (76/78 kDa) during PNS development, pro-survival CAPN1 specific degradation of PHLPP1 pathway increased in the cerebellum of developing mice, while no change was noted in CAPN1 expression [46]. It is critical to identify target proteins of calpains in myelin or axons or both during PNS development in future studies.

### 3.2. Calpains and Calpastatin during Chronic Dysmyelination in the PNS

*Tr-J* mice displaying dysmyelination are a well-established, clinically relevant model of the chronic inherited polyneuropathy CMT1E [29]. Our results demonstrate an elevation of CAPN1 and calpastatin in *Tr-J* sciatic nerves (Figure 3). Despite the increase in calpastatin, the net effect was an increase in calpain activity in the *Tr-J* sciatic nerves (Figure 3E). Similar to *Tr-J* sciatic nerves (Figure 3), activated CAPN1 and calpastatin levels tended to be higher during early developmental myelination compared to normal adult rat sciatic nerves (Figure 1). Adult *Tr-J* mouse sciatic nerves display abnormally thin myelin and short internodal length [28,31,32], similar to the histology during early PNS development in WT mice [31,45], in rats [21], and in humans [47]. Upregulation of CAPN1 and calpastatin might be associated with immature myelin sheaths and hypomyelination in the PNS. However, adult *Tr-J* mice also show pathological signs such as ongoing demyelination and remyelination [32] and macrophage infiltration [48]. The pathophysiological rationale for the changes in calpains and calpastatin in *Tr-J* sciatic nerves is unclear. Nevertheless, our findings suggest roles of calpains and calpastatin in chronic dysmyelination in the PNS. Future studies parsing the specific contributions of calpain isoforms and calpastatin in the axons, myelin, or immune cells are required for a better understanding of the pathophysiology in inherited dysmyelinating neuropathies.

### 3.3. Calpains during Acute Demyelination and Acute Axonal Degeneration in the PNS

Calpains have previously been implicated in demyelination and neurodegeneration in CNS diseases [10,49], and in axonal degeneration and node of Ranvier disruption in the PNS [11,12,13,14,15]. Additionally, a recent study demonstrated an immediate axonal cytoskeletal rearrangement in response to physiological, electrical activity in the sciatic nerve, which would be accompanied by an immediate influx of calcium ions [50]. This study implicates CAPN1 in the homeostatic plasticity of peripheral nerve structures [50]. Importantly, models for neuropathic pain demonstrated that calpain inhibitors were able to prevent the loss of myelin-associated proteins in the dorsal roots and attenuated pain-like behaviors [16,17], suggesting that calpains are also involved in myelin disruption. However, less is known about whether calpain also has a role during PNS myelin changes.

Our results from an adult acute demyelinating model further support calpains’ involvement in myelin disruption. At 6 days after a lysolecithin injection, which is the nadir of demyelination, we found significant increases in CAPN1 forms and a significant decrease in calpastatin (Figure 5). The decrease in calpastatin could be due to an increase in calpain activation, as it is a substrate of both CAPN1 and CAPN2 [51]. In addition to demyelination, there was an evident decrease in nodes of Ranvier distal to the site of the lysolecithin injection, suggesting some degree of axonal degeneration (Figure 4). To collect sufficient tissue for western blotting, we injected a greater volume of lysolecithin solution intraneurally (7 µL) than is often used in other studies. Since we saw no change in the numbers of nodes distal to the vehicle injections, the decrease in the numbers of nodes could be due to secondary axonal degeneration resulting from extensive demyelination, a common feature of many human demyelinating neuropathies [3]. However, it is unlikely that the changes in CAPN1 and calpastatin in the demyelinated areas (Figure 5) are due to the secondary axonal degeneration, because we found distinct changes of calpains and calpastatin in the degenerated nerves 6 days after a crush injury: an increase in activated CAPN1 and CAPN2 and a maintained expression of calpastatin (Figure 6). A previous sciatic nerve axotomy study showed there were no changes found in calpastatin expression while Calpain-2 levels significantly reduced over a 3-day time course after transection possibly due to autolysis after early activation, as calpain activity was found to be increased [12]. In chronic axonopathy in rat sciatic nerves induced by administration of carbon disulfide for 12 weeks, CAPN1 and CAPN2 were upregulated and calpain activity was increased [52]. Therefore, calpains may play roles in peripheral neuropathies with predominant involvement of either axons or myelin in distinct ways.

While many studies have shown a neurodegenerative effect of calpains [11,12,49], calpains may also play important roles for neuroprotection [20,46]. Using a CMT model, the Schwann cell repair phenotype has been shown to have a neuroprotective role by preventing the loss of myelinated sensory axons [53]. The transition to the repair, migratory phenotype would require a degradation of myelin and a constant remodeling of cytoskeleton, both of which contain well-established substrates of calpains, such as myelin basic protein [17,54], myelin associated glycoprotein [16,55], and spectrins [56]. In a previous study using an organotypic culture of the sciatic nerve, when calcium was deprived, there was a reduction in proliferating Schwann cells and an increase in dead and dying Schwann cells, but calcium was also required for the degeneration of axons [57]. This suggests that calcium, and possibly calpain activation, could have dual roles in response to peripheral nerve injury. Whether calpains could be playing a protective role in promoting remodeling or a destructive role in response to PNS demyelination requires further investigation. Nevertheless, our findings support the idea that modulating calpain activities could be a potential therapeutic strategy to mitigate the neurological symptoms in PNS demyelinating diseases, similar to the CNS demyelinating diseases [58,59].

### 3.4. Limitations of This Study

A technical limitation of our current study is the difficulty of sampling only the affected area of the lysolecithin injections [60]. Since lysolecithin causes a focal demyelination, our samples surely contained both demyelinating and normal sections of sciatic nerve, thus potentially obscuring changes in expression levels. Another potential limitation to the interpretation of our results is the axonal involvement in the lysolecithin-induced demyelination model. A significant decrease in nodes of Ranvier distal to the site of lysolecithin injection (Figure 4) suggests some degree of degeneration in myelinated axons. This concern is mitigated by no changes in NF-M expression between vehicle control and demyelinated nerves (Figure 5), and distinct changes of calpains and calpastatin between demyelination and axonal degeneration models (Figure 5 and Figure 6). In addition, a previous study using lysolecithin injections found a preferential loss of unmyelinated axons, while myelinated axons were spared from damage [35]. This study raises the possibility that calpains could be involved in the degeneration of unmyelinated axons and damages in non-myelinating Schwann cells in Remak bundles.

Additionally, although rodent models are commonly used to study the pathophysiology of human diseases, previous studies have demonstrated the difference between rodent and human nervous systems. For example, proteome and transcriptome studies identified multiple proteins exclusively or predominantly in human or mouse CNS myelin [61]. Even though both rat and human Schwann cells will proliferate in response to the axonal mitogen neuregulin, only the rat Schwann cells will efficiently ensheath and myelinate axons in vitro [62]. These limitations have delayed extensive experimentation in human cells. Future comparative analysis of PNS calpain-calpastatin system between rodent and human is necessary to translate our findings to humans.

## 4. Materials and Methods

### 4.1. Animal Usage

Adult SD rats were obtained from Envigo (Indianapolis, IN, USA). We obtained WT mice (C57BL/6J) and *Tr-J* mice (B6.D2-Pmp22Tr-J/J) from The Jackson Laboratory (Bar Harbor, ME, USA). Rats and mice were housed in Laboratory Animal Resources at Wright State University at 22–24 °C and 20–60% humidity under 12 h light/12 h dark conditions (lights on 7:00–19:00 h) with ad libitum access to food and water. Rats and mice were provided wood-chip bedding, toilet paper roll cardboard tubes, shredded accordion paper, and thick cotton pads for enrichment and group housed except when single housing was required for breeding purposes. Rats and mice were euthanized with an overdose of isoflurane followed by decapitation or perforation of the heart, and sciatic nerve samples were collected immediately after euthanasia. All animal procedures were approved by the Institutional Animal Care and Use Committee at Wright State University (Animal Use Protocol #1001 approved 09/08/2014, #1113 approved 08/08/2017, and #1190 approved 08/14/2020).

### 4.2. Surgical Procedures

Adult rats between P59 and P81 were anesthetized by 2% isoflurane inhalation. A small incision was made along the thigh and the left sciatic nerve was exposed. Intraneural injection of lysolecithin (Sigma-Aldrich, St. Louis, MO, USA, #L4129, L-α-Lysophosphatidylcholine) was performed as previously described with minor modifications [63]. In brief, the sciatic nerve was injected with 1% lysolecithin in sterile Locke’s solution (7 μL into tibial branch and 3 μL into peroneal branch) by using a glass micropipette broken to a tip diameter of approximately 20 μm. The Locke’s solution contained (in mM): NaCl 154, KCl 5.6, CaCl2 2, and HEPES 10, pH 7.4. For crush injury, left sciatic nerve of adult rats was firmly grasped and held for 10 s near the sciatic notch using Dumont #5 forceps. After the injection or crush, the incision was closed, and the animal was allowed to recover 6 days. For postoperative analgesia, buprenorphine (0.1 mg/kg; Reckitt Benckiser Pharmaceuticals, Richmond, VA, USA) was given subcutaneously immediately after surgery. Carprofen (5 mg/kg; Putney, Portland, ME, USA) was also given subcutaneously immediately after surgery and 24 h later.

### 4.3. Sciatic Nerve Homogenization

After euthanasia, sciatic nerve tissue was immediately collected and snap frozen in liquid nitrogen and stored at −80 °C for later testing. Snap frozen sciatic nerve samples were carefully placed in BioMasher II sterile tubes (Takara Bio, San Jose, CA, USA, #9749625002) to be homogenized in RIPA buffer (25 mM Tris HCL at pH 7.5, 150 mM NaCl, 1% Triton x100, 0.5% Deoxycholate, 0.1% SDS, 10 mM EDTA), using a pestle (BioMasher, Takara Bio, San Jose, CA, USA). Protease inhibitor (Sigma-Aldrich, St. Louis, MO, USA, #P8340) and phosphatase inhibitor (Sigma-Aldrich, St. Louis, MO, USA, #P0044; Thermo Fisher Scientific, Waltham, MA, USA, #78428) was added to the RIPA buffer. Nerves were crushed and washed in RIPA buffer with protease and phosphatase inhibitors. Samples were then placed on ice and repeatedly vortexed for about 10 min. Samples were then centrifuged at 15,000× *g* for 10 min at 4 °C in Sorvall Legend Micro 21 R Centrifuge (Thermo Fisher Scientific, Waltham, MA, USA). Supernatant was removed and aliquoted in a 0.6 mL sterile tubes. Using Pierce Bicinchoninic acid (BCA) Protein Assay (Thermo Fisher Scientific, Waltham, MA, USA, #23225), protein concentrations were measured using the Synergy H1 microplate reader (BioTek Instruments, Santa Clara, CA, USA). In a 96-well micro plate, bovine serum albumin (BSA) was used as the protein of reference (standard) for quantification of protein for each of our homogenate solutions.

### 4.4. Western Blotting

Western blotting of *Tr-J* mouse sciatic nerves was performed as previously described [64]. Modifications were made for the western blotting of development, lysolecithin, and crush rat samples as follows: samples were prepared according to the average of duplicates from BCA assay. Samples (20 μg) were denatured at 70 °C for 10 min in Novex NuPAGE LDS 4× Sample Buffer (Thermo Fisher Scientific, Waltham, MA, USA, #NP0007) with 10× reducing agent (Thermo Fisher Scientific, Waltham, MA, USA, #B0009). 2.5–5.0 μg of each rat sample was then loaded in a 4–12% gradient gel using Novex Bolt mini-gel system (Thermo Fisher Scientific, Waltham, MA, USA, #NW0412B, #B1000B, #BT000614). Using MOPS running buffer and 1 mL of Bolt antioxidant (Thermo Fisher Scientific, Waltham, MA, USA, #BT0005), the gels were run at a constant voltage of 200 V for 45 min. The gel was then transferred to nitrocellulose membrane, 0.45 µm pore size (Bio-Rad, Hercules, CA, USA, #1620115). The gels were transferred in diluted Bolt 20X transfer buffer (Thermo Fisher Scientific, Waltham, MA, USA, #BT00061), 10% methanol and 1 mL of antioxidant at a constant voltage of 13 V for 70 min. Membranes were then transferred to ultrapure MilliQ water and washed 5 min 3 times and then incubated in Azure Ponceau (VWR, Radnor, PA, USA, #10147-344) for 5 min to ensure efficiency of transfer. Following colorimetric imaging, membranes were washed in ultrapure water for 10 min 2 times. Membranes were blocked with 20 mM Tris, pH 8.0 and 0.05% (*v*/*v*) Tween 20 (TBST) containing 4% (*w*/*v*) milk for 1 h. Primary antibodies were diluted in TBST with 4% milk at 1:1000, added to the membranes and incubated overnight at 4°C. Primary antibody was washed 5 min 3 times and horseradish peroxidase (HRP) conjugated secondary antibodies (1:10,000 or 1:20,000; Jackson ImmunoResearch Laboratories, West Grove, PA, USA) were incubated for 1 h at room temperature. Signals generated by Pierce ECL Plus Western Blotting Substrate (Thermo Fisher Scientific, Waltham, MA, USA, #32132) were detected using an Azure 600 (Azure Biosystems, Dublin, CA, USA). Quantification of total band density was performed using LI-COR Image Studio software (Lincoln, NE, USA) and bands of proteins of interest were normalized to relative densities of GAPDH. Membranes were stripped with Restore stripping buffer (Thermo Fisher Scientific, Waltham, MA, USA, #21059), then washed, re-blocked and re-probed. The primary antibodies used are listed in Table 1. For between blot controls of developmental samples, a pooled sample of homogenates from 6 post weaning and adult rats was loaded into each of the gels. This common sample was then used as the reference lane for all comparisons [65]. For the lysolecithin and crush samples, a common uninjured sample homogenate was loaded into each gel as a control and used as the reference lane for all comparisons.

### 4.5. Immunofluorescent Imaging

Immunostaining of sciatic nerve sections or teased sciatic nerves was performed as described previously [66] with some modifications. In brief, upon dissection, nerves were immediately fixed in ice-cold 4% paraformaldehyde in 0.1 M phosphate buffer for 30 min, cryoprotected overnight in 20% sucrose, blocked and placed in custom-made foil blocks, and frozen in Optimal Cutting Temperature (O.C.T.) Compound (4583, TissueTek, Sakura Finetek USA, Torrance, CA, USA) and stored at −80 °C. Sciatic nerve sections (16 µm) were cut using a cryostat (HM550, Thermo Scientific, Waltham, MA, USA), collected in ice-cold 0.1 M phosphate buffer, and mounted on gelatin-coated (1%) coverslips. Some fixed sciatic nerves were gently teased apart using #7 fine forceps and spread on gelatin-coated coverslips. Sectioned or teased nerves were then blocked for 1 hr in 0.1 M phosphate buffer (pH 7.4) containing 0.3% Triton X-100 and 10% goat serum (PBTGS) then incubated overnight at 4 °C with primary antibodies diluted in PBTGS. Samples were washed three times for 10 min in PBTGS, followed by incubation in the dark with fluorescently labeled secondary antibodies for 1 h at room temperature. Alexa Fluor (594, 488) and AMCA conjugated secondary antibodies were used (Jackson ImmunoResearch Laboratories, West Grove, PA, USA). Finally, immunolabeled samples were washed in PBTGS, 0.1 M phosphate buffer, 0.05 M phosphate buffer, air-dried, and mounted onto slides using mounting medium (Product Code 71-00-16, KPL, Geithersburg, MD, USA). Images were captured on an Axio Observer Z1 with Apotome 2 fitted with a Axiocam Mrm CCD camera (ZEISS, Thornwood, NY, USA). Image analyses were performed using ZEN 2.3 software (ZEISS, Thornwood, NY, USA).

### 4.6. Quantification of Immunohistochemical Sections

Number of nodes of Ranvier in the distal tibial nerves were quantified in 4 groups: lysolecithin injected, contralateral uninjured (intact), vehicle injected, and crush injury. Six nerves in each group were examined. At least 3 sections of each nerve were examined, and 5 images from each section were randomly taken along the length of the nerve (about 15 images were obtained and analyzed from each animal). For the *Tr-J* study, 2–4 sections were used to capture about 10–15 images from each mouse. Six WT and 5 *Tr-J* nerves were examined. Counting of the nodes from a field of view (224 µm × 168 µm) was performed with Zen 2.3 software (ZEISS, Thornwood, NY, USA) by an observer blinded to experimental treatment or genotypes. Teased nerve fibers stained with the compact myelin protein DRP2 and neurofascin were used for the quantification of the myelin defect in *Tr-J* mice. Disrupted myelin sheaths were classified as either displaying disorganized DRP2 staining or having an apparent reduction in DRP2 staining. Teased sciatic nerves from 4 *Tr-J* mice were observed and about 100 myelin sheaths were analyzed from each mouse.

### 4.7. Calpain Activity Assay

A calpain activity assay was performed as described previously [67]. In brief, a Calpain Activity Assay Kit (Abcam, Cambridge, MA, USA, #ab65308) was used to quantify relative calpain enzyme activity from homogenized mouse sciatic nerves. Sciatic nerves were snap frozen in liquid nitrogen and stored at −80 °C. Sciatic nerves were pulverized with pestle (BioMasher Takara Bio, San Jose, CA, USA) and ice cold extraction buffer was added. Samples were centrifuged at 4 °C at 21,000× *g* for 5 min. Supernatants were transferred to a fresh, ice-cold tube, and protein content was quantified using a Pierce BCA Protein Assay (Thermo Fisher Scientific, Waltham, MA, USA, #23225). Changes in relative fluorescent units were detected using a Synergy H1 microplate reader (BioTek Instruments, Santa Clara, CA, USA). Relative fluorescence units for each sample were blank-subtracted, divided by the total protein loaded, and normalized to the average of vehicle controls. As controls, the *Tr-J* sciatic nerve homogenate treated with calpain inhibitor (calpeptin) was comparable to the WT nerve homogenates, providing a negative control for the assay, and active calpain was examined as a positive control, according to the manufacturer’s instruction.

### 4.8. Experimental Design and Statistical Analyses

For the *Tr-J* study, 6 WT (3 male and 3 female) and 5 *Tr-J* (2 male and 3 female) were compared (a total of 11 mice). For rat acute sciatic nerve injury models, 6 rats (3 male and 3 female) in each group (vehicle-injected, lysolecithin-injected, and crush) were used (a total of 18 rats). Comparison of the means between two groups was performed using Student’s unpaired t-test or Welch’s t-test when there was a higher variance between samples (F-test to compare two variances *p* < 0.05). One way ANOVA with Tukey’s multiple comparison test was used for 3 group comparison. All data were analyzed using Prism 9.2.0 (GraphPad, La Jolla, CA, USA) and are presented with mean ± SEM. 

## 5. Conclusions

The elevated levels of activated CAPN1 and calpastatin during early development suggest that calpains may have a role in developmental myelination in the PNS. When compared with expression at P0, the remarkably decreased adult calpastatin expression also suggests that calpain activity could be required for ongoing maintenance or remodeling of adult peripheral nerve structures including myelinated axons. We found distinct expression patterns of calpains and calpastatin between models of chronic dysmyelination, acute demyelination, and axonal degeneration. Our results suggest that the calpain-calpastatin system could be involved in a wide variety of demyelinating diseases in the PNS, such as CMT1E, the demyelinating form of Guillain-Barré syndrome, and neuropathic pain, among others. Whether these changes could be playing a protective role in promoting faster degeneration and subsequent recovery in the PNS or a detrimental role, as is often thought with calpains, requires further investigation.

## Figures and Tables

**Figure 1 ijms-23-15443-f001:**
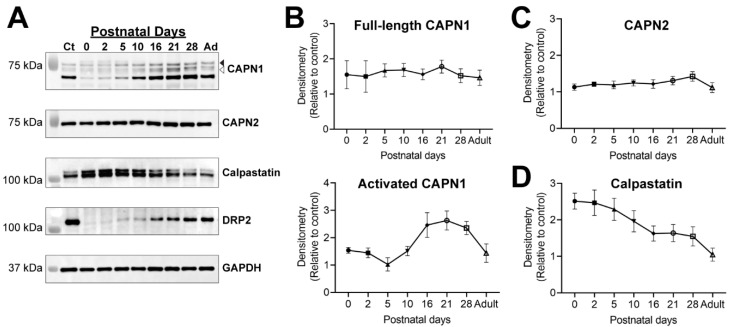
Calpain and calpastatin expression in rat sciatic nerves during development. (**A**) Immunoblotting representative images from the day of birth (P0) until adult. Relative levels of full-length CAPN1 (arrowhead at 80 kDa), amino-processed CAPN1 (open arrowhead at 76/78 kDa), CAPN2, and calpastatin were examined. A representative blot of dystrophin-related protein 2 (DRP2) is shown to demonstrate the progression of myelination during development, and GAPDH was examined as an internal loading control. Ct is a common pooled homogenate from post-weaning and adult rats for between blot comparisons. Some molecular weight markers were overlaid to illustrate band locations. (**B**–**D**) Quantitative analysis of immunoblot data during rat development. Relative density was quantified for full-length CAPN1 (80 kDa) (**B**) and amino-processed, or the activated form, CAPN1 (76/78 kDa) (**B**). Quantification of immunoblot density of CAPN2 (**C**) and of calpastatin (**D**). (**B**–**D**) Data were normalized to internal lane loading control (GAPDH) and compared with a common pooled homogenate (Ct) for between blot comparisons. Mean ± SEM graphed with *n* = 5 rats at each time point (40 rats total).

**Figure 2 ijms-23-15443-f002:**
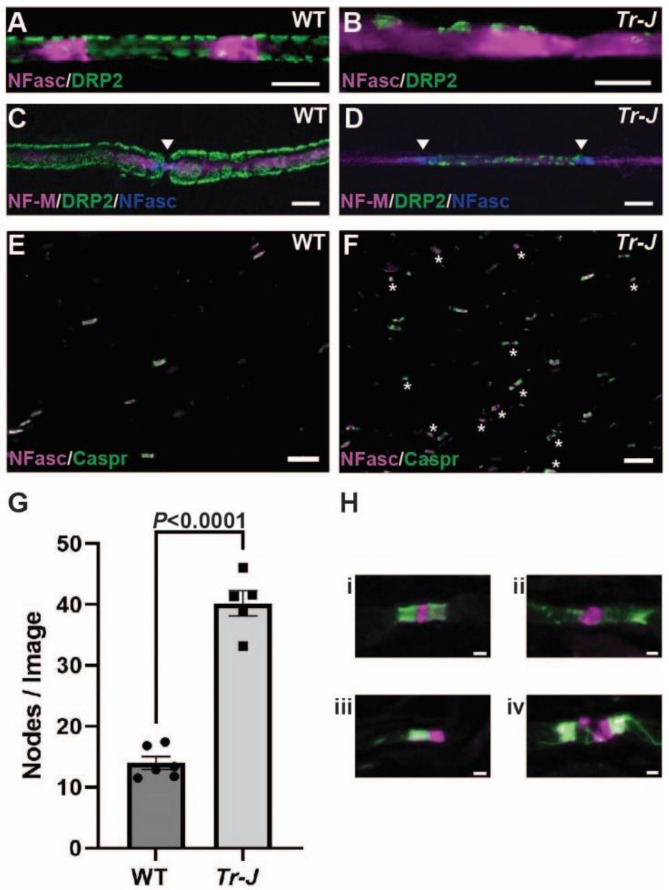
Myelin defects in *Trembler-J* mouse sciatic nerves at 8 weeks of age. (**A**–**D**) Representative immunostaining of teased sciatic nerve fibers from wild-type (WT) (**A**,**C**) and *Trembler-J* (*Tr-J*) mice (**B**,**D**). Expression and localization of Schwann cell myelin protein dystrophin-related protein 2 (DRP2, green) typically appears as a cobblestone appearance (**A**,**C**) but is reduced or altered in *Tr-J* nerves *(***B**,**D**). Neurofascin (NFasc, violet) appears to localize to Schmidt-Lanterman incisures in WT nerve (**A**), but staining is diffuse in *Tr-J* nerve (**B**). Neurofilament (NF)-M (violet) localized to the axonal cytoskeleton is preserved (**C**,**D**). Anti-neurofascin antibody used in this immunofluorescence strongly reacts to neurofascin 186 at nodal axolemma, and weakly reacts to neurofascin 155 at paranodes and Schmidt-Lanterman incisures in Schwann cells. Arrowheads indicate nodes of Ranvier (NFasc, blue) and show an abnormally short internodal length in the *Tr-J* nerve (**D**). Scale bars = 10 µm (**A**–**D**). (**E**,**F**) Representative immunostaining of sciatic nerve sections from WT (**E**) and *Tr-J* (**F**) mice for paranodal protein Caspr in green and neurofascin in violet. In (**F**), stars indicate examples of disrupted nodes of Ranvier in *Tr-J* mouse sciatic nerves. Scale bars = 20 µm (**E**,**F**). (**G**) Quantification of nodes of Ranvier density per field of view (224 µm × 168 µm) from panels (**E**,**F**) comparing sciatic nerves of WT and *Tr-J* mice. Student’s t-test was used for comparison with *n* = 6 WT and *n* = 5 *Tr-J*. Detailed results of statistical analyses are provided in the Appendix A. (**H**) Representative examples of nodes found in *Tr-J* sciatic nerve sections: (i) a normal-appearing node of Ranvier with single neurofascin cluster (violet) and two Caspr clusters (green); (ii) a single neurofascin cluster with dispersed/disorganized paranodes with Caspr staining; (iii) a single neurofascin cluster and a single Caspr cluster (hemi node); and (iv) disorganized neurofascin cluster and elongated space between Caspr clusters. Scale bars = 2 µm (H i–iv).

**Figure 3 ijms-23-15443-f003:**
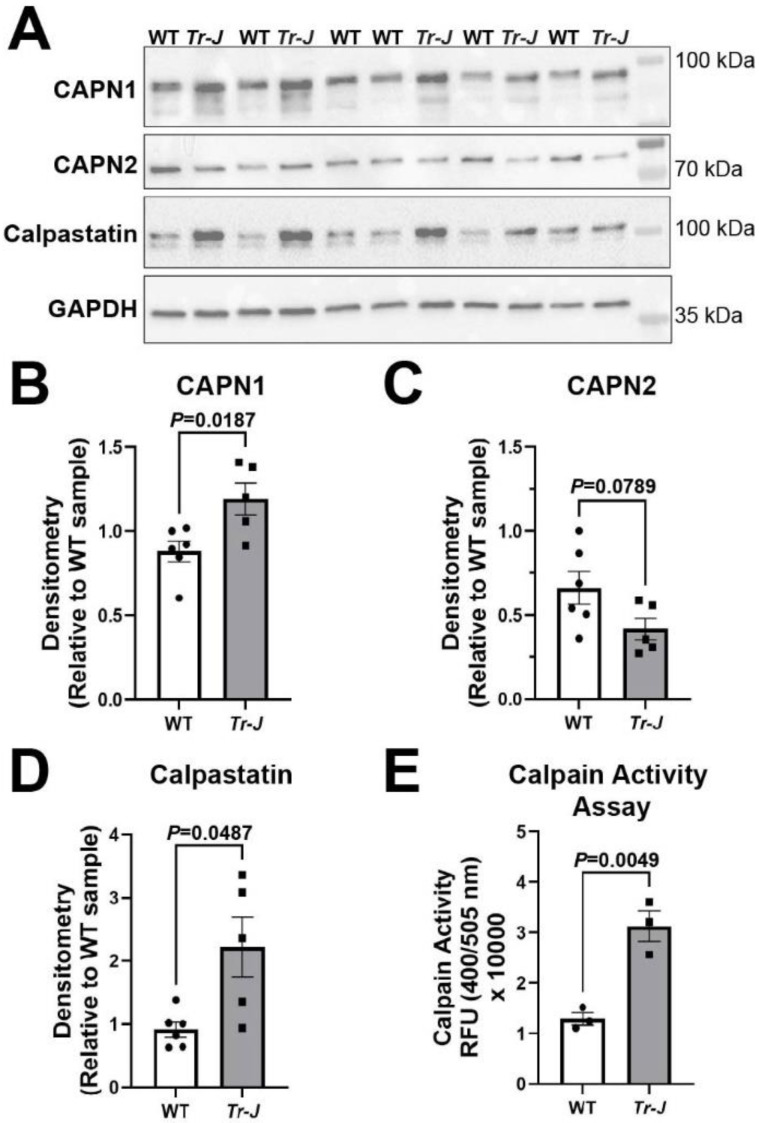
CAPN1 and calpastatin expression and total calpain activity increases in *Trembler-J* mouse sciatic nerves. (**A**) Immunoblotting from 6 wild-type (WT) and 5 *Trembler-J* (*Tr-J*) mice. Relative levels of CAPN1, CAPN2, and calpastatin were examined. GAPDH is shown as an internal loading control. Some molecular weight markers were overlaid to illustrate band locations. (**B**–**D**) Quantitative analysis of immunoblot data in WT and *Tr-J* mice. Relative density was quantified for CAPN1 (**B**), CAPN2 (**C**), and calpastatin (**D**). Since the CAPN1 full-length and amino-processed bands do not separate well in mouse tissue, the CAPN1 represents the combined full length and amino-processed activated forms. (**B**–**D**) Data are normalized to internal lane loading control (GAPDH) with mean ± SEM graphed. Student’s t-test was used for comparison of CAPN1 and CAPN2 between WT and *Tr-J* mice. Welch’s t-test was used for calpastatin analysis due to unequal variances as determined with F test (Appendix A). (**E**) Calpain substrate cleavage assay performed with sciatic nerves from WT and *Tr-J* mice. Using a Student’s t-test for comparison, a significant upregulation of calpain activity was observed in *Tr-J* mice compared to WT mice. Mean ± SEM graphed with *n* = 3 mice in each group. Detailed results of statistical analyses are provided in the Appendix A.

**Figure 4 ijms-23-15443-f004:**
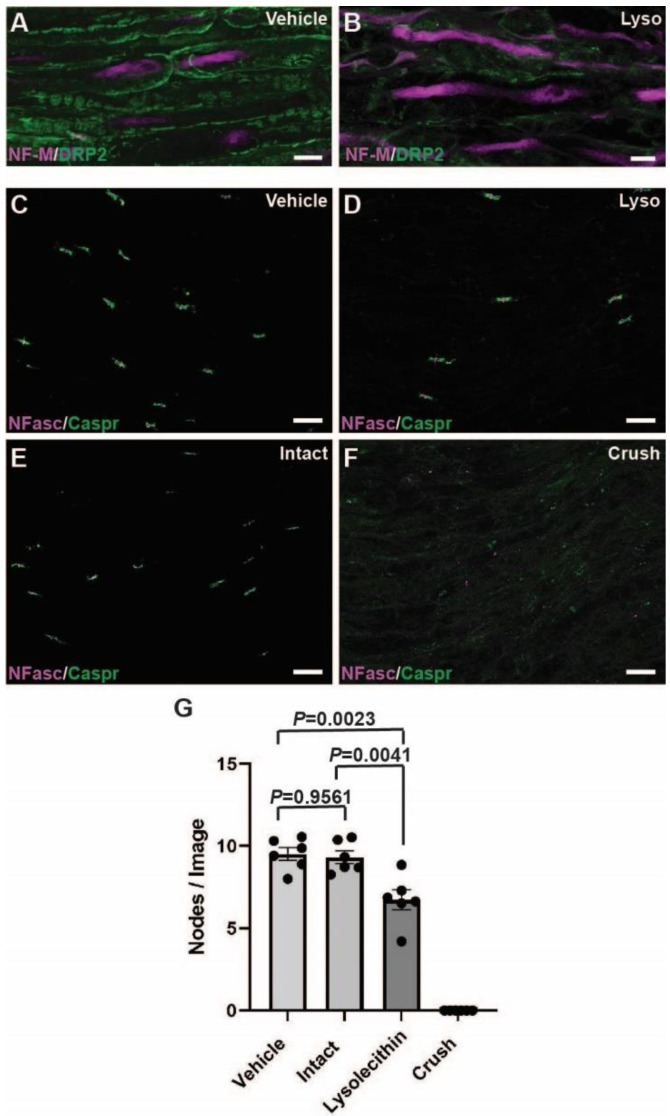
Changes in myelinated nerve fibers after intraneural lysolecithin injection or sciatic nerve crush. (**A**,**B**) Immunostaining of rat peroneal nerve sections at a vehicle injection (**A**) or lysolecithin injection (**B**) (6 days after injection). Loss of dystrophin-related protein 2 (DRP2) staining (green) in (**B**) demonstrates the loss of myelin sheaths, while the preserved neurofilament (NF)-M staining (violet) shows preserved axons. Loss of DRP2 staining was observed in sciatic nerve sections of all 6 rats used for lysolecithin injections. Scale bars = 10 µm (**A**,**B**). (**C**–**F**) Representative images of immunostaining of rat tibial nerve sections: 6 days after vehicle (**C**) or lysolecithin (**D**) injection, intact nerve (**E**), or 6 days after a crush injury (**F**). Lysolecithin-injected and contralateral intact samples were obtained from the same animals. Tibial nerves were sectioned between the knee and ankle, distal to the site of injections or crush. Structures of nodes of Ranvier were stained with neurofascin (NFasc, violet) and Caspr (green). Anti-neurofascin antibody used in this immunofluorescence strongly reacts to neurofascin 186 at nodal axolemma, and weakly reacts to neurofascin 155 at paranodes and Schmidt-Lanterman incisures in Schwann cells. Note a total disruption of nodes and paranodes after crush injury (**F**). Scale bars = 20 µm (**C**–**F**). (**G**) Quantification of nodes of Ranvier density per field of view (224 µm × 168 µm) in tibial nerves distal to site of injection or crush (panels (**C**–**F**)). A 3-group comparison among vehicle-injected nerves, intact nerves, and lysolecithin-injected nerves was performed with one way ANOVA with Tukey’s multiple comparisons test. Mean ± SEM graphed with *n* = 6 rats. The crush group was not included in statistical analysis as there were no discernible nodes of Ranvier found in the sections from the 6 animals that were analyzed. Detailed results of statistical analyses are provided in the Appendix A.

**Figure 5 ijms-23-15443-f005:**
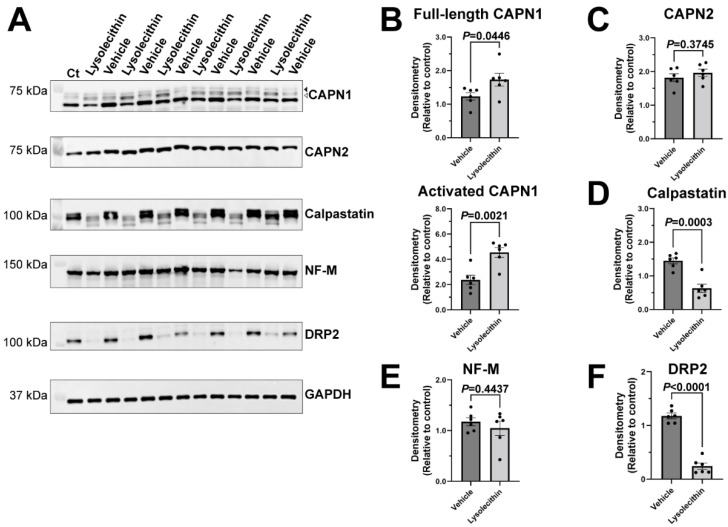
Calpain and calpastatin expression 6 days after a lysolecithin induced demyelination injury of rat sciatic nerves. (**A**) Immunoblotting images 6 days after lysolecithin induced injury. Ct represents a control homogenate from an uninjured sciatic nerve of an adult rat to be used as the reference lane. Full-length CAPN1 (arrowhead at 80 kDa), activated CAPN1 (open arrowhead at 76/78 kDa), CAPN2 and calpastatin levels were examined. Neurofilament (NF)-M expression was examined to determine the extent of axonal damage at the location of injections, and dystrophin-related protein 2 (DRP2) was examined to determine the amount of demyelination. GAPDH was examined as an internal loading control. Some molecular weight markers were overlaid to better illustrate band locations. (**B**–**F**) Quantitative analysis of immunoblot data 6 days after a lysolecithin induced injury. Relative density was quantified for the full-length CAPN1 and the activated form of CAPN1 (**B**), CAPN2 (**C**), calpastatin (**D**), NF-M (**E**), and DRP2 (**F**). Data are normalized to internal lane loading control (GAPDH). Student’s t-test was used for two group comparisons between the lysolecithin and vehicle-injected groups. Mean ± SEM graphed with *n* = 6 rats. Detailed results of statistical analyses are provided in the Appendix A.

**Figure 6 ijms-23-15443-f006:**
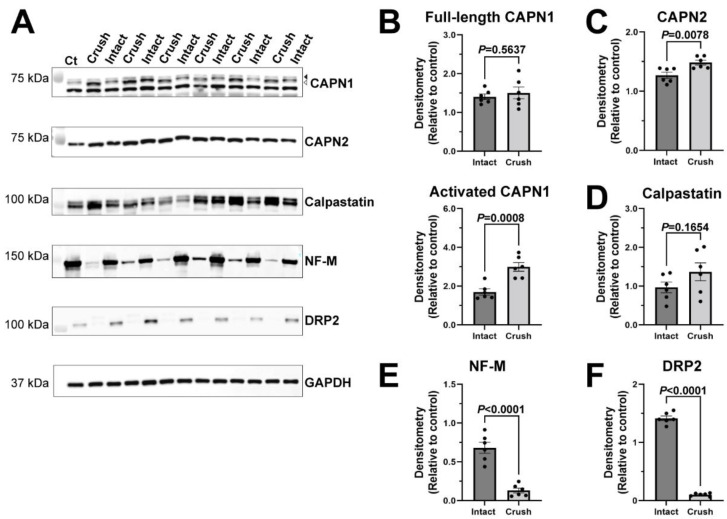
Calpain and calpastatin expression 6 days after a crush injury of rat sciatic nerves. (**A**) Immunoblotting images 6 days after a crush injury. Ct represents a control homogenate from an uninjured nerve of an adult rat to be used as the reference lane. Intact nerves are contralateral sciatic nerves to the crush injury. Full-length CAPN1 (arrowhead at 80 kDa), activated CAPN1 (open arrowhead at 76/78 kDa), CAPN2, and calpastatin levels were examined. Neurofilament (NF)-M expression was examined to determine the extent of axonal damage distal to the crush site, and dystrophin-related protein 2 (DRP2) was examined to determine the amount of myelin loss. GAPDH was examined as an internal loading control. Some molecular weight markers were overlaid to illustrate band locations. (**B**–**F**) Quantitative analysis of immunoblot data 6 days after a crush injury. Relative density was quantified for the full-length CAPN1 (80 kDa) and CAPN1 activated form (76/78 kDa) (**B**), CAPN2 (**C**), calpastatin (**D**), NF-M (**E**), and DRP2 (**F**). Data are normalized to internal lane loading control (GAPDH). Student’s t-test was used for two group comparisons between the crush-injured nerves and intact contralateral nerves (**B**–**E**). For DRP2 comparison (**F**), Welch’s t-test was used due to unequal variances according to F test (Appendix A). Mean ± SEM graphed with *n* = 6 rats. Detailed results of statistical analyses are provided in the Appendix A.

**Table 1 ijms-23-15443-t001:** Primary antibodies used for immunoblotting and/or immunofluorescent imaging.

Antigen	Dilution	Species	Manufacturer Information & RRID Citation
CAPN1	1:1000 (WB)	Rabbit	Abcam #ab28258 RRID:AB_72581
CAPN2	1:1000 (WB)	Rabbit	Cell Signaling Technology #2539 RRID:AB_2069843
Calpastatin	1:1000 (WB)	Rabbit	Cell Signaling Technology #4146 RRID:AB_2244162
DRP2	1:1000 (WB) 1:400 (IF)	Rabbit	Gift from Peter Brophy
GAPDH	1:1000 (WB)	Mouse	Enzo Life Sciences #ADI-CSA-335-E RRID:AB_2039148
Neurofascin	1:400 (IF)	Chicken	R&D Systems #AF3235 RRID:AB_10890736
Neurofilament-M	1:1000 (WB) 1:400(IF)	Mouse	Cell Signaling Technology #2838 RRID:AB_561191
AnkyrinG	1:400 (IF)	Mouse	UC Davis/NIH NeuroMab Cat. #75-146 RRID:AB_1067303
Caspr	1:400 (IF)	Rabbit	Abcam #ab3415 RRID:AB_869934

## Data Availability

Data will be available from the corresponding author on request.

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
