# Peer review of "Distinct Changes in Calpain and Calpastatin during PNS Myelination and Demyelination in Rodent Models"

_ijms, 2022, doi:10.3390/ijms232315443_

Round 1

Reviewer 1 Report

The fundamental problem of this study lies within a low sample size for the data collection.

Low sample size is not an excuse to avoid any reasons for not conducting a proper and reliable study.

In figure 1, it was indicated that “Mean±SEM graphed with n=5 rats at each time point.”, since there were 8 time points, does it mean that 40 rats were used?

Regarding statistical analysis, reviewer suggests that the results should be checked for normality of data distribution. If the data is normally distributed (assuming that n > 5 per group), parametric test of student t test should be used. Levene’s test on homogeneity of variance should be taken into consideration. Justification for statistical analysis of a small number of dataset n<5 is needed. Please refer to ARRIVE guidelines, and also other studies which could be referenced here, for example doi: 10.1016/j.neuropharm.2018.08.037.

The statistical values for parametric (t values) or non-parametric analyses should be included for all the individual results in either the result section or the figure legend.

Quantitative study is needed for results presented in figure 2. It is unclear how many animals were used for figure 2.

The details of immunohistochemical studies are missing in the methods. Particularly, the quantification (number of sections per animal and the total sections per group), randomization of sampling area, and blinded procedures of quantification with respect to individual groups should be conducted and included in the methods section.

The details of area delineation and tissue sampling for quantification, number of cells per area, etc should be provided in the method section.

In figure 3, why data were derived from 2 male and 2 female rats? These results showed huge variation of data distribution. Reviewer suggests increasing the number of animals (n=6) for only one sex.

Quantitative study is needed for figure 4.

Using an unpaired Student’s t-test for comparison in results of figure 5 is not appropriate. The sample size of n=3 mice in each group is too low for a meaningful interpretation of data.

Given that t-test was used for low sample size, first of all, authors should work out a power calculation to determine the variability and group size. A low sample size for morphological study and protein expression works would reduce the statistical power with increased Type II error skewing the data, how can one still analyze the data with parametric test? In this case, parametric test should not be used at all.

There are no statistics and p-values reported in the results section. The way the results are described and explained in the results section, is more appropriate for the discussion.

It was indicated in the result section that “These mice display tremors while walking, slowed nerve conduction and histologically display an onion-bulb appearance of peripheral nerves due  to ongoing demyelination and remyelination throughout the peripheral nerves”, still, a proper behavioral quantitative study is needed.

Reviewer 2 Report

In this study, the authors analyzed the role of the calpain/calpastatin system in myelination and demyelination in the peripheral nervous system. To this end, expression changes of calpain and calpastatin system upon 1) postnatal development, 2) lysolecithin intraneural injection-induced demyelination model, 3) Trembler-J mice were evaluated. The authors found that expression levels of CAPN1 and calpastatin were altered during PNS development, and also significantly increased in the sciatic nerves during demyelination. Through these results, the authors concluded that the calpain-calpastatin system has a putative role in demyelination diseases of peripheral nerves. Overall, a well-written, comprehensive paper with interesting results, and a meaningful approach was taken in terms of peripheral nerve biology. Despite having a relatively new and significant approach to the peripheral nerve biology, enthusiasm for research has been tempered by the following reasons:

1. The authors analyzed the role of calpain-calpastatin system through postnatal peripheral nerve development and intraneural demyelination study using SD rats and demyelination study using Trembler-J mice. In terms of comparative physiology, it would be great to mention the difference between rodent and human peripheral nerves. In particular, the number or density of nociceptors constituting peripheral nerves may have species-specific differences, indicating that rodent results may not be translated to humans.

2. The authors described that the analysis was conducted in the "development phase" through the expression analysis of CAPN1, CAPN2, Calpastatin, and DRP2 up to 28 days postnatum. In general, 4-week-old rats can be considered young or young adults. In order to analyze the role of the calpain-calpastatin system in the "developmental phase" that the author wants to talk about, isn't it necessary to analyze the pre-natal period? Rather, shouldn't the analysis of the postnatal period be viewed as changes according to "maturation or aging"? A more detailed explanation seems to be required.

3. In the lysolecithin experiment, when is the intraneural injection timing (postnatal day of rat)? Considering the changes in the expression of CAPN1 76/78 kDa and Calpastatin according to the age of rats in Figure 1, it seems necessary to mention the start time point. If injected at a young age (less than 4 weeks of age), there may be changes of the system according to maturity.

4. Changes in protein expression in Trembler-J mice are interesting. The authors mentioned that the use of calpain inhibitor (calpeptin) was comparable to the control. If so, was an improvement in the degree of demyelination observed with the use of Calpeptin? In other words, do the authors consider Calpeptin to have potential as a treatment for peripheral nerves? How's the clinical sign improvement of the animals?

5. Calpain activity assay is interesting, but more direct evidence might be necessary. If possible, the reviewer recommends to measure intracellular calcium levels during myelination/demyleination in vitro, especially if the Calpain-Calpastatin system affects calcium influx.

6. As mentioned in comment #1 and #5 above, all data of the authors originated from rat and mouse. The authors should demonstrate that the system in rodent also works in humans through validation experiments in human peripheral nerve cells.

Round 2

Reviewer 2 Report

I appreciate the great efforts that the authors have made in response to my questions and concerns. The revision clarifies all the points I raised and helps me (and hopefully readers) understand the current manuscript.